# *PDE3A* Is a Highly Expressed Therapy Target in Myxoid Liposarcoma

**DOI:** 10.3390/cancers15225308

**Published:** 2023-11-07

**Authors:** Kirsi Toivanen, Sami Kilpinen, Kalle Ojala, Nanna Merikoski, Sami Salmikangas, Mika Sampo, Tom Böhling, Harri Sihto

**Affiliations:** 1Department of Pathology, Helsinki University Hospital, University of Helsinki, 00014 Helsinki, Finland; nanna.merikoski@helsinki.fi (N.M.); sami.salmikangas@helsinki.fi (S.S.); tom.bohling@helsinki.fi (T.B.); harri.sihto@helsinki.fi (H.S.); 2Molecular and Integrative Biosciences Research Programme, University of Helsinki, 00014 Helsinki, Finland; sami.kilpinen@helsinki.fi; 3HUS Vatsakeskus, Helsinki University Hospital, PL 340, 00290 Helsinki, Finland; kalle.ojala@hus.fi; 4Department of Pathology, HUSLAB, HUS Diagnostic Center, Helsinki University Hospital, University of Helsinki, 00029 Helsinki, Finland; mika.sampo@hus.fi

**Keywords:** PDE3A, SLFN12, myxoid, liposarcoma

## Abstract

**Simple Summary:**

Liposarcomas (LPSs) are common soft-tissue sarcoma subtypes. There is an unmet clinical need for targeting LPS-subtype-specific highly expressed genes. We investigated RNA sequence data from a large clinical LPS sample series and an in silico transcriptome database consisting of 201 tissue types. We discovered that the *PDE3A* gene is highly expressed in the myxoid LPS subtype. In addition, PDE3A served as a drug target for PDE3A modulators in LPS cell lines, warranting further studies toward its usage in clinical therapy.

**Abstract:**

Liposarcomas (LPSs) are a heterogeneous group of malignancies that arise from adipose tissue. Although LPSs are among the most common soft-tissue sarcoma subtypes, precision medicine treatments are not currently available. To discover LPS-subtype-specific therapy targets, we investigated RNA sequenced transcriptomes of 131 clinical LPS tissue samples and compared the data with a transcriptome database that contained 20,218 samples from 95 healthy tissues and 106 cancerous tissue types. The identified genes were referred to the NCATS BioPlanet library with Enrichr to analyze upregulated signaling pathways. PDE3A protein expression was investigated with immunohistochemistry in 181 LPS samples, and PDE3A and SLFN12 mRNA expression with RT-qPCR were investigated in 63 LPS samples. Immunoblotting and cell viability assays were used to study LPS cell lines and their sensitivity to PDE3A modulators. We identified 97, 247, and 37 subtype-specific, highly expressed genes in dedifferentiated, myxoid, and pleomorphic LPS subtypes, respectively. Signaling pathway analysis revealed a highly activated hedgehog signaling pathway in dedifferentiated LPS, phospholipase c mediated cascade and insulin signaling in myxoid LPS, and pathways associated with cell proliferation in pleomorphic LPS. We discovered a strong association between high PDE3A expression and myxoid LPS, particularly in high-grade tumors. Moreover, myxoid LPS samples showed elevated expression levels of SLFN12 mRNA. In addition, PDE3A- and SLFN12-coexpressing LPS cell lines SA4 and GOT3 were sensitive to PDE3A modulators. Our results indicate that PDE3A modulators are promising drugs to treat myxoid LPS. Further studies are required to develop these drugs for clinical use.

## 1. Introduction

Soft-tissue sarcomas (STSs) are malignancies of mesenchymal origin that comprise approximately 1% of all cancer cases worldwide. Among the most common STS types are liposarcomas (LPSs), which arise from adipose tissue [1,2]. Due to their distinct histology and genetic background, LPSs are divided into the following four subtypes: well-differentiated LPS (WDLPS), low- and high-grade dedifferentiated LPS (DDLPS), myxoid LPS (MLPS), and pleomorphic LPS (PLPS) [1,2,3]. The 5-year overall survival rates range from 21% in high-grade DDLPS to 100% in WDLPS [1,4,5,6]. Despite varying outcomes, all LPS subtypes are usually treated similarly, with surgery combined with (neo-)adjuvant chemotherapy, radiation, or both. First-line chemotherapies used for LPS treatment include doxorubicin, which is frequently combined with ifosfamide [2,3,7,8,9,10]. A combination of docetaxel and gemcitabine is another treatment option, with survival outcomes that are similar to those resulting from doxorubicin treatment [3,7]. Two other chemotherapeutics, eribulin and trabectedin, have been approved to treat unresectable pretreated STS [2,3,10,11,12]. Eribulin treatment has shown an increased survival benefit that is greater than that of dacarbazine treatment [13]; it is somewhat more effective in DDLPS and PLPS than in MLPS [9]. In contrast, treatment with trabectedin increases progression-free survival more effectively than dacarbazine treatment, specifically in MLPS [8].

Histologically divided LPS subtypes have distinct genetic characteristics. WDLPS is a low-grade tumor that can differentiate over time into more aggressive DDLPS. However, 90% of DDLPS tumors arise de novo. Most WDLPS/DDLPS tumors have increased MDM2/CDK4 expression, due to an amplification in the chromosome 12q13-15 region [1,2]. MDM2, CDK4/6, and exportin 1 (XPO1) inhibitors are currently undergoing clinical trials for treating WD-/DDLPS [2,7,10]. Like WDLPS and DDLPS tumors, low-grade MLPS tumors can progress to a high-grade phenotype that is characterized by the presence of round tumor cells. Over 90% of MLPS tumors contain a fusion gene (*FUS-DDIT3* or *EWSR1-DDIT3*) that drives tumorigenesis [1,2,14]. Moreover, MLPS tumors frequently have upregulation of the IGF1R/PI3K/Akt signaling pathway, driven by the fusion protein; targeting these pathway alterations with PI3K and mTOR inhibitors is being investigated [1,15,16,17,18]. Highly aggressive PLPS is the rarest LPS subtype and accounts for approximately 5% of all LPSs. Due to its rarity, collecting large sample series has been challenging and no definite genetic aberrations or specific therapy targets of PLPS have been identified [1,19]. In summary, LPSs are a group of molecularly, histologically, and clinically heterogeneous tumors, and there is an unmet clinical need for targeting LPS subtype-specific highly expressed genes.

Phosphodiesterases (PDEs) constitute a superfamily of enzymes (PDE1–PDE11) whose main function is to regulate the signaling of secondary messengers cAMP and cGMP by hydrolyzing 3′,5′-cyclic nucleotides to their 5′-monophosphate form. Tissue-specific expression of various PDE isoforms enables cellular function regulation in a cell-type specific manner. Aberrant expression of PDE isoforms is linked to many diseases [20,21]. PDE3A-targeting compounds have recently been described as inhibiting tumor growth in mice inoculated with PDE3A-expressing cervical cancer HeLa [22], neuroglioma H4 [23], and melanoma SK-MEL-3 cells [24], and in patient-derived gastrointestinal stromal tumor (GIST) xenograft models [25]. These compounds act like a molecular glue and form a complex between PDE3A and Schlafen 12 (SLFN12), leading to cell-cycle arrest or cell death [22,23,26,27,28,29,30]. While the precise downstream signaling pathways remain unclear, the formed complex stabilizes SLFN12 and its RNase activity, resulting in protein synthesis inhibition [26,27,31].

We sought to discover novel LPS-subtype-specific oncogenes that could serve as precision medicine targets. RNA-Seq transcriptome analysis of LPS patient tissue samples were first compared to transcriptomes of other cancerous and healthy tissue samples, resulting in ranked lists of highly expressed genes in three LPS subtypes and their enrichment in biological signaling pathways. PDE3A expression was high in MLPS, and its expression frequency in other STS subtypes and its potential role as a therapy target in LPS were further investigated.

## 2. Materials and Methods

### 2.1. Patient Cohort and Selection Criteria

Patients who were diagnosed with LPS from 1987 to 2017 in the Helsinki University Hospital (HUH) were identified. Clinical data were extracted from the patient registry, and hematoxylin and eosin-stained tissue slides and corresponding formalin-fixed, paraffin-embedded (FFPE) tissue blocks were extracted from the archives of the Department of Pathology. Histological classifications of tumors were reviewed by Mika Sampo, one of the authors, based on the 2020 WHO classification of soft-tissue tumors [1]. A total of 190 patients with primary tumor samples were identified for the retrospective study series. Two samples were excluded due to missing clinical data (MLPSs), four samples were excluded due to a paucity of tumor tissue, and three samples were excluded due to technical problems in immunohistochemistry (WDLPS, DDLPS, MLPS). Thus, a total of 181 samples were included in the series. The median follow-up time for survival in the series was 4.7 years (the range was from 0.3 to 20.3).

RNA was extracted from 186 LPS FFPE tissue samples, including samples that were excluded from immunohistochemistry (IHC) analysis and two samples for which histology was reviewed but clinical data were not available. The RNA samples were subjected to RNA sequencing, resulting in sequences from 152 samples. Of these, 131 transcriptomes passed the quality control analysis and were further investigated for LPS-subtype-specifically overexpressed genes. Of note, all WDLPS samples (*n* = 10) were excluded from the analyses, due to low RNA yield from sequencing.

In addition, a previously collected series of 543 STS samples were investigated by using IHC and tissue microarray (TMA) from the following subtypes: fibrosarcoma, GIST, LPS, leiomyosarcoma, malignant peripheral nerve sheath tumor, myxofibrosarcoma, sarcoma with unspecified histotype, and synovial sarcoma [32]. TMA blocks included two to three 1-mm diameter core punches from each tumor sample. Tissue samples were retrieved from the archives of the Department of Pathology, HUH. TMAs were generated with an automated tissue microarrayer (TMA Grand Master, 3DHISTECH Ltd., Budapest, Hungary) in the Biobank Helsinki.

### 2.2. RNA Extraction and Sequencing

Tissue sections (2 × 10 µm) were collected in 2-mL microfuge tubes (Cat. No. 72.693.005, Sarstedt, NRW, Nümbrecht, Germany) and deparaffinized with 160 µL of deparaffinization solution (Cat. No. 939018, QIAGEN, NRW, Hilden, Germany). If tumor cells were estimated to present >50% of all cells in a tissue section, a representative tumor area was scraped to the tube from the slide. Samples were mixed by vortexing and, subsequently, centrifuged and incubated at 56 °C for 3 min. One hundred and twenty µL of buffer PKD (Cat. No. 1034963, QIAGEN) was added, mixed by vortexing, and centrifuged for 13,000× *g* for 1 min. Ten µL of proteinase K (Cat. No. 19133, QIAGEN) was added to the lower liquid phase of the samples. After a 1 h incubation at 56 °C in a tube shaker, samples were centrifuged and RNA was extracted with QIAsymphony RNA Kit (Cat. No. 931636, QIAGEN) using a QIAsymphony^®^ SP instrument (QIAGEN). RNA quality and concentration were measured with a 2100 Bioanalyzer instrument (Agilent, Santa Clara, CA, USA).

RNA sequencing was conducted, as described earlier [33]. Shortly after adding 1 µL of ERCC RNA spike-in control (1:1000) to each sample, an RNA sequencing library was prepared with a QuantSeq 3′ mRNA Library kit (Lexogen, Vienna, Austria) following the manufacturer’s instructions, as previously described. Sequencing was performed with a HiSeq 2500 System (Illumina, San Diego, CA, USA) using a high-throughput mode and v4 chemistry in three batches that included samples from different LPS subtypes. Sequences were processed with the data analysis pipeline, the BlueBee Genomics platform (BlueBee Holding BV, NB, Rijswijk, The Netherlands), following the manufacturer’s instructions.

### 2.3. Processing of Sequencing Data

Htseq-count files from the Bluebee sequencing process were read into R [34] (version 4.1.2) and matched to clinical data per sample. Gene annotation was retrieved from AnnotationHub (snapshotDate: 20 October 2021). Data were imported into a DGEList object by using edgeR package [35] (version 3.36.0). Genes were further filtered by the filterByExpr function (default parameters) from edgeR. log2 counts per million were calculated with the cpm function (default parameters) from edgeR. Additional sample quality control was performed by excluding samples with a median logcount, with deviation >10% from the median logcount over all samples (21 samples were excluded, resulting in 131 samples for downstream analysis). The final analysis included RNA sequence data from 53 DDLPS samples, 54 MLPS samples, and 24 PLPS samples.

### 2.4. BAM File QC Metrics

QC metrics for the BAM files produced by Bluebee pipeline were additionally calculated by parallel [36] run of RNA-SeQC 2 [37] with summarising results with MultiQC [38]. Genome annotation file and version used in the QC calculations was Homo_sapiens. GRCh38.104.genes.gtf (collapsed to gene level), also available from project Github page. The sequencing coverage and quality statistics of each sample are presented in Appendix A.

### 2.5. Transcriptome Reference Data Fetching and Processing

Data from the newest version of in silico transcriptomic (IST; [39]) database (MediSapiens Ltd., Helsinki, Finland) were fetched into R by using Dplyr extension for SQlite. We calculated the mean expression for each gene (excluding genes with <6 datapoints) per each healthy anatomical tissue type (*n* = 95) and per each cancerous tissue type (*n* = 106). For each tissue type (healthy and cancerous), we then chose the top 500 expressing genes. The final IST reference gene list was calculated by choosing a unique list of genes, in which each gene did not appear in more than one of the healthy-tissue top 500 lists and in more than three of the cancer-tissue top 500 lists. This final IST reference list contained a total of 2754 genes and was used as an exclusion list, to eliminate non-LPS-specific genes from the LPS-subtype differential-gene-expression analyses.

### 2.6. Identification of DEG between LPS Subtypes

A differential expression test of genes for the LPS subtypes was performed, using the estimateDisp and exactTest functions from edgeR, with default parameters. Differential-expression genes (DEGs) were defined as genes with a positive logFC > 1.5 (with *p* value < 0.05) between one subtype and all other subtypes, resulting in three lists of DEGs. Altogether, we identified 97 DEGs between the DDLPS subtype and the other two subtypes, 247 DEGs between the MLPS subtype and the other two subtypes, and 37 DEGs between the PLPS subtype and the other two subtypes, after using the IST-derived exclusion lists (see above).

### 2.7. LPS-Subtype-Specific Pathway Analysis

Upregulated signaling pathways in the three LPS subtypes were described with all of the identified subtype-specific DEGs, using Enrichr [40] and the NCATS BioPlanet 2019 library [41]. The scatter plots and the bar charts with their related genes were visualized via Appyters (https://appyters.maayanlab.cloud/#/). The pathway search was conducted on 30 June 2023.

### 2.8. Quantitative Real-Time PCR of LPS Tissue RNA

PDE3A and SLFN12 mRNA levels were measured from 63 randomly selected FFPE LPS samples, excluding WDLPS samples. RNA extracted from the FFPE block derived from the GIST882 cell line was used as a reference sample for PDE3A and SLFN12 expression levels. cDNA was synthesized from RNA with SuperScript™ IV VILO™ Master Mix (Cat. No. 11756500, Invitrogen™, Thermo Fisher Scientific, Waltham, MA, USA). Primers and hybridization probes (Universal ProbeLibrary Set, Human Probes #1–#90, Cat. No. 04683633001 Roche, Basel, Switzerland) were designed using ProbeFinder version 2.53 for Human. The primers were as follows: PDE3A, F: 5′-AAA GAC AAG CTT GCT ATT CCA AA-3′, R: 5′-GTG GAA GAA ACT CGT CTC AAC A-3′; SLFN12, F: 5′-CTT TGT TCA ACA CGC CAA GA-3′, R: 5′-ATG CAG TGT CCA AGC AGA AA-3′; and YWHAZ, F: 5′-CGT TAC TTG GCT GAG GTT GC-3′, R: 5′-TGC TTG TTG TGA CTG ATC GAC-3′. Real-time qPCR was performed with the LightCycler^®^ 480 Probes Master Kit (Cat. No. 04887301001, Roche) and the CFX96 Touch Real-Time PCR detection system (Bio-Rad, Hercules, CA, USA), under the following conditions: 95 °C for 10 min; 45 cycles of 95 °C for 30 s, 60°C for 30 s, and 72 °C for 45 s with plate read (amplification); 72 °C for 7 min; 65 °C to 95 °C (increment 0.01 °C s^−1^, melt curve). PCR was performed in triplicate for each sample. Relative PDE3A and SLFN12 mRNA expression were calculated using the ΔΔCt method and normalized to YWHAZ expression.

### 2.9. Immunohistochemical Staining

Slides with 3.5-µm tissue sections were incubated at 56 °C for 1 h prior to deparaffinization with Tissue-Tek DRS 2000 Automated Slide Stainer (Sakura Finetek Japan, Tokyo, Japan). After blocking endogenous peroxidase with hydrogen peroxide, antigen retrieval was performed with EnVision FLEX Target Retrieval Solution Low pH (pH 6, Cat. No. K8005, Agilent) in a decloaking chamber (Biocare Medical, Pacheco, CA, USA) at 110 °C for 3 min. The slides were then incubated with polyclonal rabbit anti-PDE3A (1:100, Cat. No. HPA014492, Sigma-Aldrich, St. Louis, MO, USA) in WellMed BrightDiluent for 30 min (Cat. No. BD09-500, ImmunoLogic, Duiven, The Netherlands). WellMed BrightVision (one component detection system, Goat Anti-Rabbit IgG HRP, Cat. No. DPVR500HRP, ImmunoLogic) was used for antibody detection and the slides were stained with an ImmPACT^®^ DAB Peroxidase (HRP) substrate kit (Cat. No. SK-4105, Vector Laboratories, Burlingame, CA, USA). The tissues were counterstained with hematoxylin (Mayer’s, Cat. No. S3309, Agilent).

The slides were scanned with Pannoramic 250 Flash III (3DHISTECH) in the Biobank Helsinki and analyzed with a slide viewer (3DHISTECH). Interstitial cells of Cajal from healthy colon tissue were used as a positive control for PDE3A staining. The intensity of PDE3A staining on TMA was scored as 0 (no staining in tumor cells), 1 (weak), 2 (moderate), or 3 (strong). Due to a heterogeneous staining pattern in some of the samples, PDE3A expression was evaluated in whole-tissue sections in the LPS series, based on H-scores (staining intensity (0–3) × percentage of positive cells). For statistical analysis, the results were further divided into quartiles (0: 0–75; 1: 76–150; 2: 151–225; 3: 226–300) or into two groups (low: 0–150, and high: 151–300), when comparing the differences between LPS subtypes.

### 2.10. Cell Culture

GOT3 (RRID:CVCL_M819), MLS1765-92 (RRID:CVCL_S817), and MLS402-91 (RRID:CVCL_S813) cell lines were established and kindly provided by Prof. Pierre Åman (University of Gothenburg, Sweden); GIST882 (RRID:CVCL_7044) and LPS141 (RRID:CVCL_M823) cell lines were kindly provided by Dr. Jonathan Fletcher (Harvard Medical School, Boston, MA, USA); and the SA4 (RRID:CVCL_8910) cell line was kindly provided by Dr. Kjetil Boye (Oslo University Hospital, Oslo, Norway). The 93T449 (RRID:CVCL_U614) (Cat. No. CRL-3043), 94T778 (RRID:CVCL_U613) (Cat. No. CRL-3044) and SW872 (RRID:CVCL_1730) (Cat. No. HTB-92) cell lines were purchased from ATCC (Manassas, VA, USA). All cell lines used in this study were authenticated using short-tandem-repeat (STR) profiling in the genotyping lab of the FIMM Technology Centre. GIST882 was verified as containing a known KIT exon 13 p.K642E mutation. The SA4 cell line was verified by Western blot as not being HeLa-contaminated, where the HPV18 E7 protein was not detected. All cell lines tested negative for mycoplasma and were cultured in a humidified, 5% CO_2_ atmosphere at 37 °C. The cell lines were cultured in Gibco™ RPMI Medium 1640 (Cat. No. 11530586, Life Technologies, Carlsbad, CA, USA) with Pen Strep Glut and 5% (MLS402-91 and MLS1765-92), 10% (SA4, GOT3, 93T449, and 94T778), or 20% (GIST882, LPS141 and SW872) FBS.

### 2.11. Western Blotting

The cells were lysed in M-PER™ Mammalian Protein Extraction Reagent (Cat. No. 78501, Thermo Fisher Scientific) with HALT™ protease and phosphatase inhibitor cocktails (Cat. No. 78429 and Cat. No. 78420, Thermo Fisher Scientific). Denatured lysates were subjected to SDS-PAGE, using Mini-PROTEAN^®^ TGX™ Precast Gels (Cat. No. 456-1034, Bio-Rad), and blotted to PVDF membranes (Mini Format, Cat. No. 1704156, Bio-Rad). After blocking, the membranes were incubated overnight in primary antibodies at 4 °C. Polyclonal goat anti-rabbit IgG (1:10,000, Cat. No. 111-035-003, Jackson ImmunoResearch, West Grove, PA, USA) was used as a secondary antibody. Antigens were detected with SuperSignal™ West Pico PLUS chemiluminescent substrate (Cat. No. 34580, Thermo Fisher Scientific) on autoradiography films and quantitated using Fiji ImageJ 1.53 (64-bit). Antibodies used for Western blotting were polyclonal rabbit anti-PDE3A (1:1000, Cat. No. HPA014492, Sigma-Aldrich) and monoclonal rabbit anti-SLFN12 (1:500, Cat. No. ab234418, Abcam, Cambridge, UK). Target protein expression was normalized to polyclonal rabbit anti-cytoskeletal actin (1:150 000, Cat. No. A300-491A, Bethyl Laboratories, Montgomery, TX, USA).

### 2.12. Cell Viability and Cytotoxicity Experiments

The following drugs were used in the experiments: anagrelide hydrochloride (CAS No. 58579-51-4, Cat. No. 735554, Lancrix, Shanghai, China), BAY 2666605 (CAS No. 2275774-60-0, Cat. No. HY-145924, MedChemExpress, Monmouth Junction, NJ, USA), cilostazol (CAS No. 73963-72-1, Cat. No. C0737, Sigma–Aldrich), and DNMDP (CAS No. 328104-79-6, Cat. No. HY-W028690, MedChemExpress). All of the drugs were dissolved in dimethyl sulfoxide (DMSO). A corresponding DMSO concentration in cell media was used as the control treatment.

A total of 10,000 GIST882 cells, 3000 SA4 cells, and 1800 GOT3 cells were seeded per well on a Falcon^®^ 96-Well White/Clear bottom microplate (Cat. No. 353377, CORNING, Corning, NY, USA) and allowed to attach for 24 h before the addition of drugs. The number of cells plated in wells differed between cell lines to attain 70–80% confluency in the control treatment on the final measuring day. Cell viabilities were measured with CellTiter-Glo^®^ Luminescent Cell Viability Assay (Cat. No. G7572, Promega, Madison, WI, USA), following the manufacturer’s instructions. Luminescence was recorded with a Hidex Sense microplate reader (Hidex Oy, Turku, Finland). Background values from media were excluded from the results. Eight well replicates per treatment were always performed. The results presented were obtained from two separate experiments. Luminescence measurements were divided by the average of control treatment, indicating a relative cell viability change percentage.

### 2.13. Statistical Analysis

Statistical analyses were performed using IBM SPSS Statistics 28.0.0.0 (Chicago, IL, USA). *p* values < 0.05 were considered statistically significant. Cross tabulations were analyzed using a Pearson χ^2^ test or a Fisher–Freeman–Halton test. Correlations between PDE3A staining with age at time of diagnosis and tumor size were examined using the Mann–Whitney U Test. Correlations between PDE3A expression and overall survival or metastasis-free survival were estimated with the Kaplan–Meier method and compared using the univariate Cox proportional hazard model. Survival was calculated from the date of diagnosis to the date of death or the date of the first detected metastases, excluding patients who were without any events at the end of follow-up. The correlation between the PDE3A H-score and RT-qPCR expression was analyzed using Spearman’s rank-order correlation, and the correlation between PDE3A and SLFN12 mRNA expression was analyzed with Pearson correlation. Statistical significance was analyzed using the Kruskall–Wallis test to compare PDE3A and SLFN12 mRNA expression levels between the LPS subtypes and gender. Differences in cell viability across treatments were analyzed using a two-tailed unpaired *t*-test.

## 3. Results

### 3.1. Identification of Potentially Targetable Subtype-Specific Genes and Pathways

A comparison between transcriptomes of 53 DDLPS samples, 54 MLPS samples, and 24 PLPS samples and the IST transcriptome database identified 97 highly expressed genes specific in DDLPS, 247 highly expressed genes specific in MLPS, and 37 highly expressed genes specific in PLPS (Appendix A). The top 25 upregulated genes in each subtype, with the corresponding logFC (fold change) and *p* values, are shown in Table 1. Genes related to *CDK4* and *MDM2* co-amplification, *B4GALNT1*, *TSPAN31, YEATS4, HMGA2, FRS2,* and *GLI1* have been described in WDLPS/DDLPS and emerged in our DDLPS gene list [42,43,44,45,46]. *SHANK2* and *CTAG2* emerged in the MLPS-specific list and *AXL* emerged in the PLPS-specific list; these genes have previously been described as upregulated in these subtypes [47,48,49]. For emerging oncogenic genes that have been characterized in other cancers, we identified *PAPPA* [50], *FBN2* [51], and *METTL1* [52] in DDLPS, *POTEE* [53] and *LBX1* [54] in MLPS, and *CDCA2* [55] and *TROAP* [56] in PLPS as differentially expressed genes.

Figure 1 shows the top 10 enriched pathways, with their overlapping genes, in three LPS subtypes. The most important pathways in DDLPS were the hedgehog signaling pathway and the calcium-activated potassium channel, which were previously described as being involved in many cancers [57,58,59]. The two most upregulated genes identified in DDLPS, *GLI1* and *HHIP*, are related to the hedgehog signaling pathways. The MLPS-specific DEGs displayed upregulated phospholipase C-mediated cascade and insulin signaling pathways, which are also recognized pathways that are altered in cancer [60,61]. The pathways identified in PLPS formed an apparent cluster and are involved in M phase, DNA replication, kinesins, and cell cycle.

### 3.2. PDE3A mRNA and Protein Expression Is Frequent in LPS

Phosphodiesterase 3A (*PDE3A*) emerged in the MLPS gene list (Appendix A) and is related to upregulated insulin signaling (Figure 1). PDE3A protein is highly expressed in cardiovascular tissues [21] and has been identified as a promising drug target in GISTs [25,62]. Therefore, we decided to further investigate its role in LPS. IHC staining for PDE3A in a TMA series consisting of nine different STS types revealed expression in nearly half of the samples (Table 2). Examples illustrating the staining intensities in different STS types are shown in Figure 2. Consistent with previous studies, PDE3A was most frequently expressed in GISTs [25,62]. GIST samples also exhibited PDE3A expression that was significantly higher than it was in other STS subtypes. Other STS types that most often exhibited moderate or high PDE3A expression (staining intensity ≥2) were leiomyosarcoma (39.0%), malignant peripheral nerve sheath tumors (25.0%), not-otherwise-specified sarcoma (21.2%), and LPS (19.5%).

### 3.3. Elevated PDE3A and SLFN12 Expression Is Typical for MLPS

The transcriptome data (Appendix A) indicated that PDE3A expression was higher in the MLPS subtype than in various other tissue types or the two other LPS subtypes. Therefore, we investigated 181 LPS tissue sections with specified histology, using IHC. Consistent with the transcriptome data, elevated PDE3A expression was associated with the MLPS subtype, but also with male gender (Table 3) (*p* < 0.05). Associations between PDE3A expression and tumor site, age at time of diagnosis, tumor size, metastasis at diagnosis, overall survival (hazard ratio = 0.853, 95% confidence interval = 0.479–1.519, *p* = 0.588), or metastasis-free survival (hazard ratio = 0.905, 95% confidence interval = 0.374–2.190, *p* = 0.824) were not observed.

PDE3A expression was also analyzed by dividing the samples into quartiles, based on H-scores (Figure 3A). Examples of PDE3A immunostaining intensity levels in LPS tissues are shown in Figure 3B. Interestingly, in comparison to low-grade MLPS, high PDE3A expression was particularly associated with high-grade MLPS (*p* = 0.032) (Figure 3A). Higher PDE3A expression in MLPS was also seen in mRNA levels (Appendix A), and a positive correlation between mRNA and protein expression was observed (Spearman’s ρ = 0.423, *p* < 0.001) (Appendix A).

The cytotoxic effect of PDE3A modulators is based on their ability to induce complex formation between PDE3A and SLFN12, and drug sensitivity positively correlates with expression levels of the two proteins [22,23,26,27,28,29,30,63]. We examined SLFN12 mRNA expression in 62 LPS samples and found it to be expressed at higher levels in MLPS samples than in DDLPS and PLPS samples (Figure 3C) (Kruskal–Wallis test, *p* = 0.012). In addition, there was a correlation between SLFN12 and PDE3A mRNA expression levels (Pearson correlation coefficient = 0.362, *p* = 0.004); almost all of the samples (11/12) that exhibited equal or greater PDE3A and SLFN12 mRNA expression than a PDE3A modulator-sensitive cell line were MLPS samples (Figure 3D). No association between male gender and PDE3A was observed in the mRNA levels (Kruskal–Wallis test, *p* = 0.389). Similarly, no association was detected between SLFN12 mRNA expression and male gender (Kruskal–Wallis test, *p* = 0.880).

### 3.4. PDE3A- and SLFN12-Coexpressing LPS Cell Lines Are Sensitive to PDE3A Modulators

Finally, we examined whether PDE3A modulators may have efficacy in treating LPS. PDE3A and SLFN12 protein expression were first evaluated in eight LPS cell lines (Figure 4A). PDE3A- and SLFN12-coexpression was detected in two of the eight LPS cell lines, SA4 and GOT3, and in the GIST882 control cell line. We then measured, with CellTiterGlo, the effect of three PDE3A modulators (anagrelide, BAY 2666605, and DNMDP) and one PDE3A enzyme inhibitor (cilostazol) on cell viability in these cell lines. Cell viabilities decreased with all PDE3A modulators in all three cell lines (two-tailed unpaired *t*-test, *p* < 0.001) (Figure 4B). No response was observed with cilostazol, indicating that PDE3A-and SLFN12-positive LPS cell lines are affected via the PDE3A–SLFN12 complex and not by enzyme inhibition.

## 4. Discussion

Efficacious subtype-specific targeted therapies are currently not available for LPSs. In this study, we sought to discover novel genes that are specifically overexpressed in LPS subtypes, via a large-scale transcriptome analysis. We identified 381 genes, many of which may have a role in tumorigenesis or cancer biology pathways and, therefore, may serve as therapeutic targets in LPSs. As expected, the analysis yielded some known LPS-subtype-specific genes, but also revealed multiple genes that are linked to other cancer types. As many of these genes have been described as possibly targetable—for example *PAPPA* in breast cancer [50] and *TROAP* in glioma [56]—the discovery of their LPS-subtype-specificity is worth considering for further investigation.

In addition to identifying individual therapeutic targets, we were interested in examining signaling pathways associated with the overexpressed genes to describe the LPS subtypes and to understand the possible mechanisms of tumorigenesis. The emerging hedgehog signaling in the DDLPS-specific pathway analysis is a compelling discovery, as targeting sonic hedgehog signaling and its key factor, GLI1, has been studied in other sarcomas [57]. Moreover, we identified *GLI1* as the most upregulated gene in DDLPS, and previous studies have also shown high expression of two *GLI* homologs, *GLI1* and *GLI2*, in DDLPS [43,46,64]. MLPS tumors have frequently been described as having an activated IGF1R/PI3K/Akt signaling pathway that is induced by the FUS-DDIT3 oncoprotein [18]. Interestingly, a highly homologous pathway, the insulin signaling pathway, was observed as the second-most-upregulated pathway in our analysis, after the phospholipase C-mediated cascade. Almost all of the upregulated pathways in PLPS were involved in DNA replication and cell proliferation. Aberrations in cell-division-associated pathways aligns with morphological features of PLPS, which frequently exhibit a high number of mitoses and multinucleated giant cells [19]. Further studies should be conducted to investigate whether these pathways can be targeted—for example, with cyclin-dependent kinase inhibitors.

Although *PDE3A* was only ranked 124th on the MLPS DEG list, it caught our attention because it has been recognized in GISTs [25,62] and described as potentially targetable with drugs, called PDE3A modulators, that are cytotoxic to PDE3A- and SLFN12-coexpressing cell lines [23,24,25,26,27,28,29,30]. The high PDE3A expression in GISTs and cardiovascular tissue might explain why it ranked relatively low in our analysis. In addition to confirming the association between PDE3A expression and the myxoid subtype, via IHC staining, high and frequent PDE3A expression was specifically seen in the high-grade MLPS samples with a round cell phenotype. High PDE3A expression, in protein level, was also associated with male gender, but no other connections with clinical factors were observed. Our pathway analysis identified PDE3A as part of the insulin signaling pathway, which shares significant similarities with the IGF1R/PI3K/Akt pathway, driven by the commonly observed FUS-DDIT3 fusion protein in MLPS [18]. However, further research is needed to determine if elevated PDE3A expression is associated with the fusion oncogene-driven pathway alterations. The MLPS samples also exhibited higher SLFN12 mRNA expression levels than those of the DDLPS and PLPS samples. Interestingly, most of the samples that presented PDE3A and SLFN12 mRNA expression levels that were higher than those of the GIST882 cell line belonged to the myxoid subtype. This finding is intriguing and warrants further efficacy studies of PDE3A modulators in LPS in vivo models, as these compounds may lead to an MLPS-specific targeted therapy.

The study had certain limitations. As frozen tissue samples for RNA extraction were not available, we used RNA isolated from FFPE samples, in which RNA is known to be degraded. However, in a previous study, we demonstrated that the transcriptomic data derived from FFPE provide profiles that are similar to those of RNA data from fresh-frozen tissue in Merkel cell carcinoma [33]. We were unable to find any PDE3A-positive MLPS cell lines for the study. Instead, we used two PDE3A- and SLFN12-positive LPS cell lines, GOT3 classified as a WDLPS, and SA4 with unspecified histology, which showed responses to PDE3A modulators. Therefore, an association of frequent and high PDE3A expression and the efficacy of PDE3A modulators in MLPS was not confirmed in vitro. However, other studies have shown that PDE3A- and SLFN12-expressing cell lines and xenografts of various cancer types are sensitive to PDE3A modulators, suggesting that the mechanism behind the cytotoxic effect is universal and not cell-type dependent [22,25,26,27,29,30].

In conclusion, our findings suggest that LPSs, specifically MLPS tumors that coexpress PDE3A and SLFN12, may be candidates for PDE3A modulator treatment. Further in vivo experiments and optimization of PDE3A and SLFN12 diagnostics should be conducted to identify suitable patients for therapy, ultimately enabling efficient patient stratification and precision medicine in the future.

## 5. Conclusions

In summary, we identified LPS-subtype-specific highly expressed genes, which could potentially be targeted with precision medicine. These genes are associated with the hedgehog signaling pathway in DDLPS, phospholipase C, and insulin signaling pathways in MLPS and cell proliferation pathways in PLPS. Through in vitro experiments, we observed elevated PDE3A expression correlating with high-grade myxoid samples and discovered that two LPS cell lines were responsive to PDE3A modulators. Further research is required to determine whether PDE3A modulators could potentially be used as precision medicine in treating MLPS—specifically, in high-grade tumors.

## Figures and Tables

**Figure 1 cancers-15-05308-f001:**
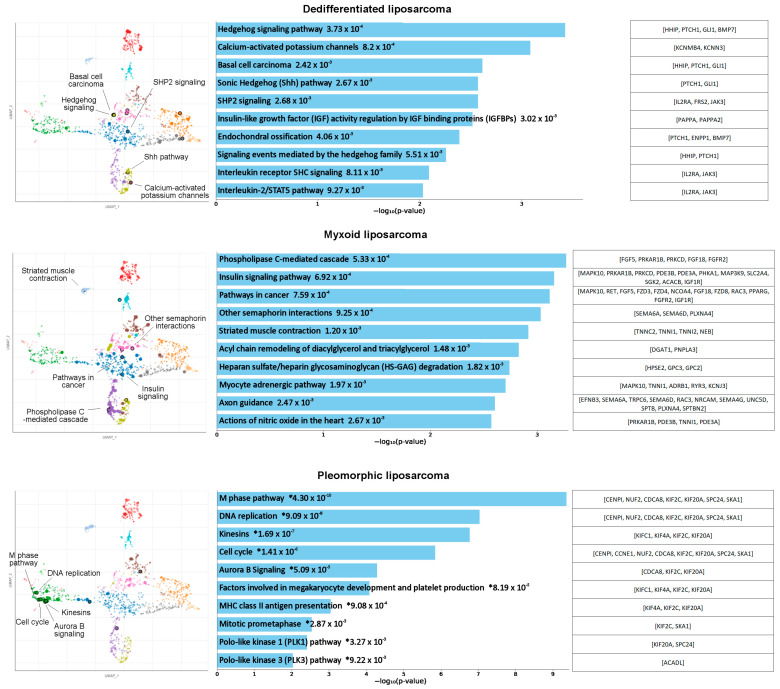
LPS-subtype-specific pathway analysis. The pathway analysis was performed with NIH pathway data source Bioplanet 2019, utilizing the lists of upregulated genes in LPS subtypes, yielding the top 10 upregulated pathways shown in bar charts, with their corresponding *p* values. Clustering of the top 5 signaling pathways is shown in the scatter plots, where pathways with more similar gene sets are positioned closer together. The overlapping genes to each pathway are presented on the right. * The term also has a significant adjusted *p* value (<0.05).

**Figure 2 cancers-15-05308-f002:**
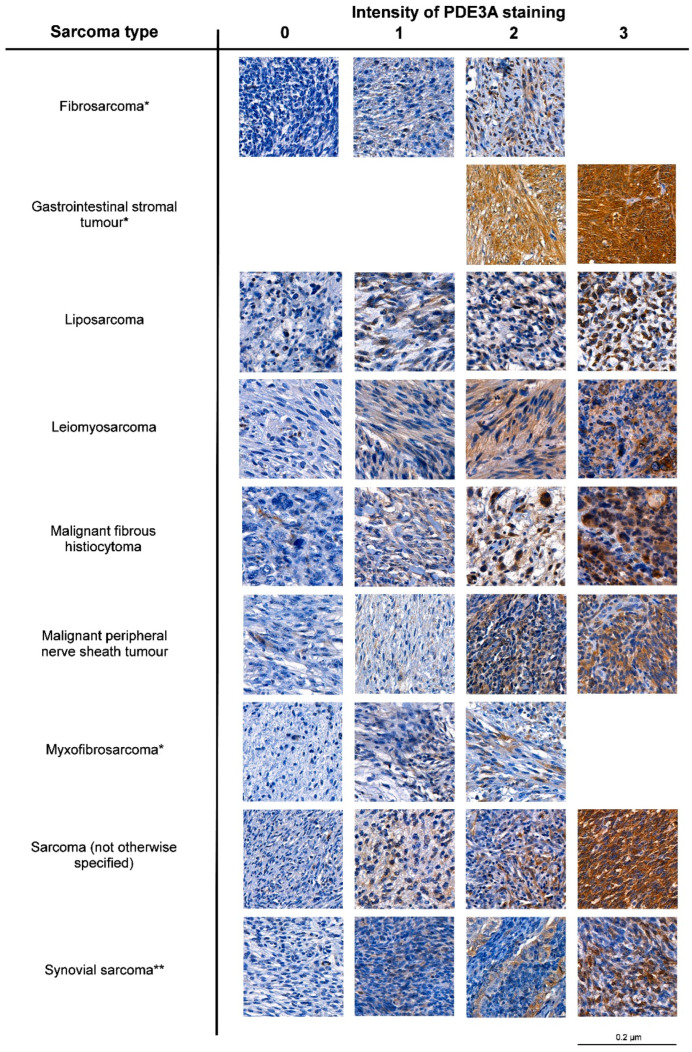
PDE3A staining in nine soft-tissue sarcoma types. Images were acquired from scanned stained TMA slides that were viewed with CaseViewer. Scale bar: 0.2 µm. * Not all staining intensities appeared. ** In some cases, only gland-like epithelial structures of the tumor were stained.

**Figure 3 cancers-15-05308-f003:**
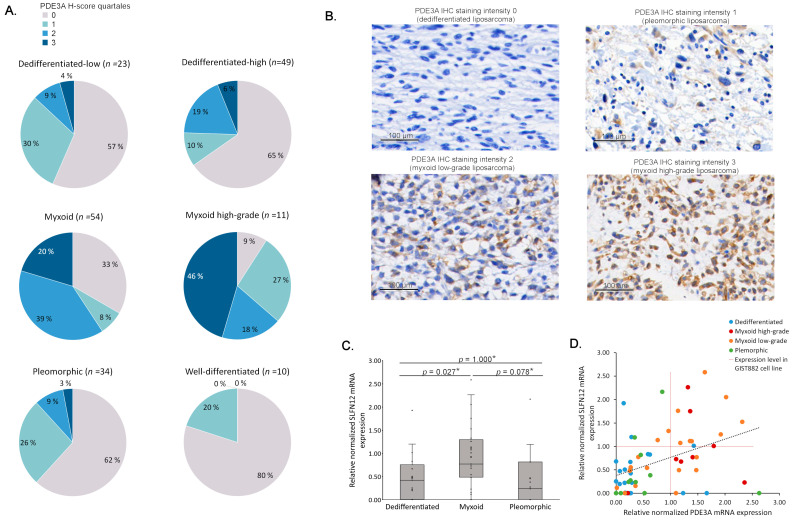
PDE3A and SLFN12 expression in LPS subtypes. (**A**) PDE3A H-score results in LPS subtypes. Myxoid samples showed higher PDE3A expression than other samples (crosstabulation and Pearson χ^2^ test, asymp. sig., two-sided, *p* < 0.001), when low (0 and 1) and high (2 and 3) H-score classes were compared in three subtypes (WDLPS/DDLPS, MLPS, and PLPS). (**B**) Example of PDE3A IHC staining intensities in different LPS subtypes. (**C**) Normalized SLFN12 mRNA expression of 62 LPS samples (21 DDLPS, 29 MLPS, and 12 PLPS). Data are presented as median relative to that of GIST882 cell line in FFPE (=1). A statistically significant difference in SLFN12 mRNA expression was observed between subtypes with the Kruskal–Wallis test (*p* = 0.012). * The significance values in pairwise comparisons of subtypes have been Bonferroni corrected. (**D**) A correlation between SLFN12 and PDE3A mRNA expression levels was observed (Pearson correlation coefficient = 0.362, *p* = 0.004). Data are presented as relative to that of the GIST882 cell line in FFPE (red lines =1).

**Figure 4 cancers-15-05308-f004:**
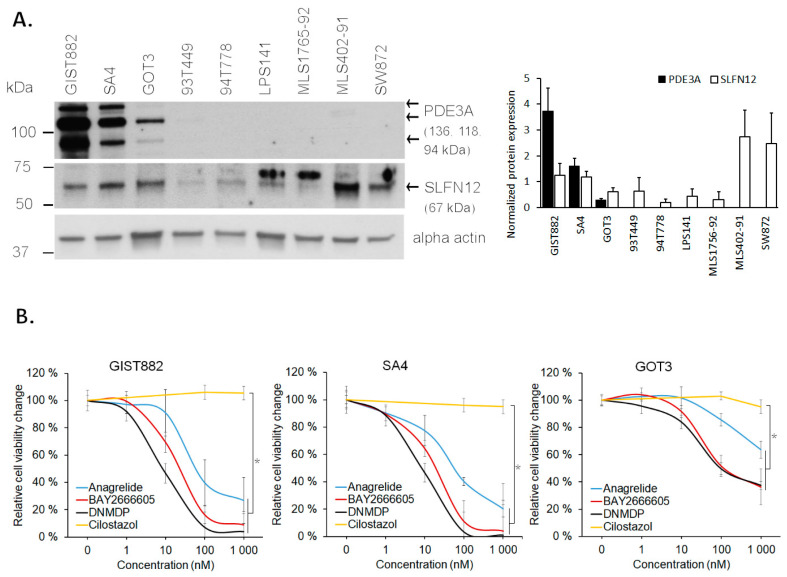
Two PDE3A- and SLFN12-coexpressing LPS cell lines are sensitive to PDE3A modulators. (**A**) PDE3A expression and SLFN12 expression were detected with Western blot in two out of eight LPS cell lines; SA4 and GOT3. GIST882 was used as a positive control in the analysis. Original Western blots are presented in Appendix A. (**B**) PDE3A modulators reduced cell viabilities of GIST882, SA4, and GOT3, while the PDE3A inhibitor, cilostazol, did not. Cilostazol was measured in two concentrations, 100 nM and 1000 nM. * *p* < 0.001.

**Table 1 cancers-15-05308-t001:** Top 25 overly expressed genes in dedifferentiated, myxoid, and pleomorphic LPS subtypes.

	Dedifferentiated			Myxoid			Pleomorphic		
	Symbol	logFC	*p* Value	Symbol	logFC	*p* Value	Symbol	logFC	*p* Value
1	*GLI1*	4.019	8.5674 × 10^−12^	*POTEE*	6.724	2.4549 × 10^−34^	*TROAP*	2.374	3.1991 × 10^−12^
2	*HHIP*	3.981	2.8013 × 10^−11^	*GNAT3*	6.010	2.6398 × 10^−24^	*HAS1*	2.117	0.000052
3	*B4 GALNT1*	3.322	3.8753 × 10^−13^	*SHANK2*	5.408	4.2321 × 10^−34^	*SPC24*	2.022	7.1793 × 10^−8^
4	*PAPPA2*	2.856	5.1590 × 10^−7^	*SLC17A8*	5.106	7.1407 × 10^−20^	*CDCA2*	1.958	8.3866 × 10^−9^
5	*LRRC4B*	2.828	4.0311 × 10^−7^	*LBX1*	4.757	3.4411 × 10^−25^	*CNKSR2*	1.948	0.00015
6	*FBN2*	2.731	3.983 × 10^−11^	*UNC5D*	4.666	6.53067 × 10^−24^	*UPK3BL1*	1.944	0.0015
7	*ENTPD2*	2.672	1.5089 × 10^−8^	*RASGEF1C*	4.491	1.2908 × 10^−22^	*TMEM170B*	1.858	0.000035
8	*PTCH1*	2.566	5.0399 × 10^−11^	*SPATA22*	4.387	4.4755 × 10^−19^	*DEPDC1*	1.853	1.1343 × 10^−6^
9	*METTL1*	2.530	9.3422 × 10^−14^	*KLHDC8A*	4.303	8.3055 × 10^−19^	*KIF2C*	1.835	7.3270 × 10^−7^
10	*COLGALT2*	2.499	3.7948 × 10^−9^	*AMN*	4.159	1.3435 × 10^−20^	*DLGAP5*	1.827	1.0800 × 10^−8^
11	*PI15*	2.458	1.2388 × 10^−7^	*POU3F3*	4.087	5.5783 × 10^−18^	*C18orf54*	1.824	2.1967 × 10^−7^
12	*AVIL*	2.457	2.0786 × 10^−14^	*CTAG2*	4.074	7.8181 × 10^−12^	*CCNE1*	1.799	0.000078
13	*GALNT17*	2.333	1.3743 × 10^−8^	*NPW*	4.034	3.4685 × 10^−16^	*TNFAIP8L3*	1.741	0.00057
14	*HMGA2*	2.308	0.000032	*CSMD1*	3.980	4.3107 × 10^−19^	*SLC6A8*	1.722	0.00080
15	*IRAK3*	2.307	1.3883 × 10^−14^	*ADAMTS19*	3.970	3.8458 × 10^−16^	*ACAN*	1.714	0.0017
16	*PAPPA*	2.263	3.7620 × 10^−9^	*SOX1*	3.928	8.7166 × 10^−15^	*KIF14*	1.707	5.75845 × 10^−7^
17	*SLC35E3*	2.240	6.6762 × 10^−15^	*SIM1*	3.927	4.2415 × 10^−23^	*KIF20A*	1.691	0.00013
18	*ATP23*	2.231	4.6716 × 10^−9^	*GIPR*	3.855	2.2921 × 10^−18^	*RFX8*	1.682	0.00035
19	*YEATS4*	2.226	7.2431 × 10^−13^	*NELL1*	3.803	1.4580 × 10^−15^	*CDCA5*	1.681	7.7032 × 10^−6^
20	*PRRT2*	2.195	6.2120 × 10^−10^	*DPP10*	3.783	3.4701 × 10^−14^	*DPP4*	1.622	0.00039
21	*GRIN2D*	2.188	1.8108 × 10^−7^	*MYH15*	3.777	8.4681 × 10^−19^	*NUF2*	1.621	1.2021 × 10^−6^
22	*NTN1*	2.168	4.9020 × 10^−8^	*COL23A1*	3.745	1.7877 × 10^−18^	*CEP55*	1.617	0.000017
23	*TUBB3*	2.142	1.5008 × 10^−6^	*TTPA*	3.743	1.6634 × 10^−13^	*ACADL*	1.615	0.0019
24	*C1QL1*	2.102	8.84882 × 10^−6^	*KCNJ3*	3.683	2.1530 × 10^−15^	*RNF112*	1.612	0.00061
25	*MDM2*	2.100	1.1237 × 10^−11^	*BMPR1B*	3.542	4.8406 × 10^−19^	*GFRA2*	1.606	0.000083

**Table 2 cancers-15-05308-t002:** PDE3A IHC staining results of soft-tissue sarcoma sample series.

Sarcoma Type		PDE3A Intensity				Total
		0	1	2	3	
Fibrosarcoma	N	7	2	1	0	10
	%	70	20	10	0	
Gastrointestinal stromal tumor	N	0	0	8	19	27
	%	0	0	29.6	70.4	
Liposarcoma	N	46	24	14	3	87
	%	52.9	27.6	16.1	3.4	
Leiomyosarcoma	N	19	17	17	6	59
	%	32.2	28.8	28.8	10.2	
Malignant fibrohistiocytoma	N	133	67	19	3	222
	%	59.9	30.2	8.6	1.4	
Malignant peripheral nerve sheath tumor	N	11	4	3	2	20
	%	55	20	15	10	
Myxofibrosarcoma	N	27	14	7	0	48
	%	56.3	29.2	14.6	0	
Sarcoma (not otherwise specified)	N	14	12	5	2	33
	%	42.2	36.4	15.2	6.1	
Synovial sarcoma	N	29	4	3	1	37
	%	78.4	10.8	8.1	2.7	
Total	N	286	144	77	36	543
	%	52.7	26.5	14.2	6.6	

**Table 3 cancers-15-05308-t003:** Association between PDE3A expression and clinicopathological factors.

	Total Cases	PDE3A H-Score		*p* Value
		Low	High	
All cases	181	123 (68.0%)	58 (32.0%)	
Sex				0.016
Male	95	57 (60.0%)	38 (40.0%)	
Female	86	66 (76.7%)	20 (23.3%)	
Histological subtype				<0.001 *
Well-differentiated	10	10 (100%)	0 (0%)	
Dedifferentiated	72	57 (79.2%)	15 (20.8%)	
Myxoid	65	26 (40.0%)	39 (60.0%)	
Pleomorphic	34	30 (88.2%)	4 (11.8%)	
Tumor site				0.060 *
Limb	83	50 (60.2%)	33 (39.8%)	
Retroperitoneal	48	39 (81.3%)	9 (18.8%)	
Trunk	36	25 (69.4%)	11 (30.6%)	
Abdomen	11	8 (72.7%)	3 (27.3%)	
Head	3	1 (33.3%)	2 (66.7%)	
Age at the time of diagnosis				0.070
Median (range in years)	59	60 (20−92)	54 (22−88)	
Tumor size				0.532
Median (range in cm^3^)	10	10 (1.5−40)	9.25 (2.2−30)	
Data not available		17	8	
Metastasis at diagnosis				0.271
Present	8	4 (2.2%)	4 (2.2%)	
Not present	173	119 (65.7%)	54 (29.8%)	

* Fisher–Freeman–Halton test.

## Data Availability

The IST database was in-licensed from MediSapiens Ltd. (Helsinki, Finland). Clinical registry data and associated RNA sequence data used in the study are stored and managed by the Rare Cancers Research Group at the Department of Pathology, University of Helsinki, Helsinki, Finland. The data cannot be shared publicly due to sensitive patient information collected from hospital medical records and the Finnish Cancer Registry (https://cancerregistry.fi/ (accessed on 1 December 2018)) and national legislation concerning the use of personal data (the Act on Secondary Use of Health and Social Data (552/2019). The data will be shared upon reasonable request via an application made to the Ethics Committee of Helsinki University Hospital (https://www.hus.fi/tutkimus-ja-opetus/tutkijan-ohjeet/eettisen-lausunnon-hakeminen). Other data are available from the corresponding authors upon reasonable request. The datasets and computer code produced in this study are available in the following databases: R code used to process bulk-RNAseq data, generate the IST database gene exclusion list, and perform DEG tests of liposarcoma samples: Github (https://github.com/Rare-Cancers-Research-Group/PDE3A-targeting-therapies-in-liposarcoma).

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
