# Peer review of "PDE3A Is a Highly Expressed Therapy Target in Myxoid Liposarcoma"

_cancers, 2023, doi:10.3390/cancers15225308_

Round 1

Reviewer 1 Report

Comments and Suggestions for Authors

In this manuscript, the authors collected a large cohort of liposarcoma FFPE samples (dedifferentiated, myxoid, and pleomorphic) and conducted transcriptome sequencing on those that met quality criteria. This is by far one of the largest liposarcoma cohorts to be characterized in this way and is important as these tumors can be heterogeneous within each subtype. The use of FFPE is a limitation, however the large number of samples may mitigate this effect. Using the gene expression profiles, the authors identified genes that distinguished each liposarcoma subtype from the others. One of these genes, PDE3A, is highly expressed in myxoid liposarcoma as compared to dedifferentiated and pleomorphic liposarcoma. Immunohistochemistry results confirm this finding. Treating cells with PDE3A modulators may decrease overall cell viability. Overall, the scientific concepts have potential. However, there are major concerns with the differential gene expression analysis and the use of SA4 to validate the findings for PDE3A.

Major comments

1.       For quality assurances, can the authors comment on what was done to prevent batch effects? The ideal situation would be to submit samples from different liposarcoma subtypes into each sequencing batch. If this was not done, computational batch effect removal may have to be conducted, but very carefully so as not to remove biological differences among the subtypes. For these difficulties in batch effects, the single-sample gene set enrichment analysis is effective for comparing gene expression levels across batches.

2.       According to the Human Protein Atlas, PDE3A RNA expression is on the lower side in adipose tissue, but not as high as heart. It would be helpful to see if these levels in normal tissues can be related to those found in the liposarcoma in this study here to help provide context as to the levels of expression seen here in liposarcoma (how high is “high”?). This would be helpful to understand whether PDE3A is a viable therapeutic target.

3.       Why are MDM2 and CDK4 not listed as one of the top differentially expressed genes (higher in dedifferentiated liposarcoma versus myxoid and pleomorphic)? The absence of these genes in the results suggest that the analysis is flawed and should be redone. Please see the microarray results here where MDM2 and CDK4 are more highly expressed in DDLPS than in pleomorphic liposarcoma in Figure 2 of this publication: PMID: 17638873 DOI: 10.1158/0008-5472.CAN-07-0584. Also please relate any of the findings here to those in this same paper.

4.       Are any of the myxoid-specific genes, especially PDEA4, downstream of the fusion FUS/EWSR1-DDIT3? This would be helpful to validate the findings.

5.       According to the Cellosaurus website, SA4 is a problematic cell line (https://www.cellosaurus.org/CVCL_8910) and these results should not be included in the manuscript.

6.       Is there increased apoptosis in GOT3 and GIST882 after treatment with PDEA4 modulators?

Minor comments

1.       In line 302, HDMGA4 is not a known gene. Please correct this.

2.       Please color the dots in Figure S1 according to the liposarcoma subtype for clarity.

Comments on the Quality of English Language

There are some parts that could be edited so that the manuscript has more formal language, but overall, it is well-written.

Reviewer 2 Report

Comments and Suggestions for Authors

In this paper, the authors have described that high-level expression of PDE3A is associated with myxoid liposarcoma (MLPS), and SLFN12 expression is also elevated in MLPS.  Furthermore, PDE3A modulators, which induce complex formation between PDE3A and SLFN12, are suggested to be promising drugs for treatment of myxoid LPS. 

These results contain interesting points, and may be important for development of a novel therapy for myxoid LPS.

The reviewer has some question as follows that the authors should be addressed.

1. On lines 329-330 in Results, the authors wrote that " Phosphodiesterase 3A (PDE3A) emerged in the MLPS-specific gene list and is related 329 to upregulated insulin signaling."

The authors should clearly indicate what the MLPS-specific gene list is.

2. On lines 347-348 in Results, the authors wrote that "The transcriptome data indicated that PDE3A expression is specific for the MLPS subtype." 

The authors should clearly indicate what the transcriptome data is.

In addition, the reviewer suspects that the expression of PDE3A is not specific to MLPS, as there are cases such as GIST. Please explain this.

3. On lines 376-378 in Results, the authors wrote that " We examined SLFN12 mRNA expression in 62 LPS samples and observed that SLFN12 expression was higher in MLPS than in DDLPS and PLPS samples (Figure 3C)."

Have the authors not confirmed the correlation between the expression levels of SLFN12 and PDE3A in each sample?

4. On lines 401-402 in Discussion, the authors wrote that "We identified 381 overexpressed genes that may have a role in tumorigenesis and therefore may serve as therapy targets.

However, the 381 genes must have simply been highly expressed in DDLPS, MLPS or PLPS.  

Is there any evidence that all the genes are involved in tumorigenesis? 

Round 2

Reviewer 1 Report

Comments and Suggestions for Authors

The authors have addressed the majority of comments and have improved the manuscript. The therapeutic benefit of PDE3A inhibitor will likely benefit only the small subset of high grade myxoid liposarcoma patients with the highest levels of PDE3A.

1. Since gender may play a role in the , can the authors also code the data according to gender? Fig. 3D and Suppl. Fig. S1B
